# Comparing the 30-Day Mortality for Hip Fractures in Patients with and without COVID-19: An Updated Meta-Analysis

**DOI:** 10.3390/jpm13040669

**Published:** 2023-04-15

**Authors:** Sojune Hwang, Chiwon Ahn, Moonho Won

**Affiliations:** Department of Emergency Medicine, College of Medicine, Chung-Ang University, Seoul 06974, Republic of Korea; junee327@cauhs.or.kr (S.H.); wonmh0922@cauhs.or.kr (M.W.)

**Keywords:** hip fracture, COVID-19, pandemic, mortality

## Abstract

We conducted an updated meta-analysis to evaluate the 30-day mortality of hip fractures during the COVID-19 pandemic and assess mortality rates by country. We systematically searched Medline, EMBASE, and the Cochrane Library up to November 2022 for studies on the 30-day mortality of hip fractures during the pandemic. Two reviewers used the Newcastle–Ottawa tool to independently assess the methodological quality of the included studies. We conducted a meta-analysis and systematic review including 40 eligible studies with 17,753 patients with hip fractures, including 2280 patients with COVID-19 (12.8%). The overall 30-day mortality rate for hip fractures during the pandemic was 12.6% from published studies. The 30-day mortality of patients with hip fractures who had COVID-19 was significantly higher than those without COVID-19 (OR, 7.10; 95% CI, 5.51–9.15; I^2^ = 57%). The hip fracture mortality rate increased during the pandemic and varied by country, with the highest rates found in Europe, particularly the United Kingdom (UK) and Spain. COVID-19 may have contributed to the increased 30-day mortality rate in hip fracture patients. The mortality rate of hip fracture in patients without COVID-19 did not change during the pandemic.

## 1. Introduction

The COVID-19 pandemic has significantly impacted the healthcare system and redirected many of its resources [1]. There was a shortage of information about the disease, and many healthcare systems faced collapse in the early stages of the pandemic [2,3,4]. This caused a gap in care for non-COVID-19 diseases, including delays in diagnosis and treatment due to additional screening processes in hospitals and emergency care systems [5]. The pandemic also impacted emergency medical services for diseases such as acute myocardial infarction, stroke, out-of-hospital cardiac arrest, and sepsis [3,4,6,7,8,9]. During the pandemic, the epidemiological characteristics of known diseases have also changed. In fact, for out-of-hospital cardiac arrest, there was an increase in arrests occurring at home and a decrease in the frequency of shockable rhythms. These changes were associated with an increase in out-of-hospital cardiac arrest cases and longer transport times to hospitals following COVID infection [8,9]. Furthermore, various surgical diseases have either contributed to the increase in the frequency of surgical decisions during the pandemic or led to a rise in complications due to delays in surgery. For example, patients with appendicitis faced delays in surgery and increased complications [10]. Social distancing and self-isolation measures during the pandemic further exacerbated these issues.

Hip fractures, a growing public health concern among the elderly, have high mortality rates [11]. Hip fracture management indicates the quality of care for elderly patients and how trauma services are functioning [12]. Despite the decreased number of trauma patients due to reduced activity during the pandemic, the number of hip fractures in elderly individuals was unchanged [13]. The social distancing, self-isolation, and limited public medical services during the pandemic made caring for the elderly more difficult. COVID-19 screening processes may delay surgery for patients with hip fractures [14]. The pandemic also increases the risk of death for those with hip fractures, with previous meta-analyses reporting higher mortality rates for patients with both hip fractures and COVID-19 compared to those with hip fractures alone [15,16].

These meta-analyses evaluated the effect of COVID-19 infection on the outcomes for patients with hip fractures and showed significant results. Even after that, numerous studies investigated the effect of COVID-19 on the outcomes of hip fracture. At this point, we needed to analyze the changed results compared to the results of previous studies. Therefore, we conducted an updated meta-analysis on the 30-day mortality of hip fractures for individuals with and without COVID-19 and also analyzed the 30-day mortality of hip fractures during the pandemic based on published cases.

## 2. Materials and Methods

### 2.1. Reporting Guidelines and Protocol Registration

This study complied with the Preferred Reporting Items for Systematic Reviews and Meta-analyses and the Meta-analysis of Observational Studies in Epidemiology Guidelines for Reporting Information from Observational Studies [17,18]. We prospectively registered with the PROSPERO registry (CRD42022385443).

### 2.2. Eligibility Criteria

We applied the Population, Intervention, Comparison, and Outcome (PICO) clinical question. We performed a literature search and selected eligible studies. The study outcomes were then evaluated in a meta-analysis. The PICO questions were as follows: population (P) = all adult patients with hip fractures visiting the emergency room regardless conduction of operation; exposure (I) = COVID-19 infection; comparator (C) = non-infection; outcome (O) = 30-day mortality.

### 2.3. Search Strategy

Two reviewers systematically searched several electronic databases (Medline via OVID interface, Embase, and Cochrane Library) for studies on the outcomes of adult hip fracture patients with COVID-19 infection compared to those with no infection. The search terms were “Coronavirus” or “COVID-19” or “SARS-CoV” or “2019-nCoV” or “Severe Acute Respiratory Syndrome”, and “Hip Fractures” or “Femoral Fractures” or “femoral shaft” or “femur shaft” or “periprosthetic” or “femur neck” or “trochanteric” or “intracapsular”. We summarize the detailed search strategy for each database in Appendix A.

### 2.4. Study Selection

Two reviewers independently screened the title, abstract, and type of each identified article, excluding irrelevant studies. First, we eliminated duplicate studies in which the titles, authors, and publication years were the same. We then excluded all reviews, case reports, case series, editorials, comments, or meta-analyses; animal studies; research with irrelevant study populations; and those with inappropriate controls. A third reviewer intervened if the two reviewers disagreed about a study, and differences were discussed until a consensus was achieved. Finally, we included studies that evaluated the outcomes of patients with hip fractures during the COVID-19 pandemic and compared them to those reported before the pandemic and studies on adult populations over 18 years of age. We also excluded studies that (1) included patients aged less than 18 years, (2) provided no comparisons or outcomes, and (3) were non-original articles. In addition, we included fracture patients with or without surgery. We subsequently reviewed the full text of potentially relevant articles that met the inclusion criteria.2.5. Data Extraction

The two reviewers independently extracted the following information from the included studies: (1) publication details (author and year), (2) study type and settings, (3) patient population (region, number of participating center(s), number of patients, and patient demographics and comorbidities), (4) the rate of operation, (5) the length of hospital stay, and (6) 30-day mortality. Discrepancies between reviewers were resolved by consensus.

### 2.5. Quality Assessment in Individual Studies

We assessed the risk of bias in each study with the Newcastle–Ottawa Quality Assessment for Cohort Studies tool [19]. Each study was assigned stars for three domains: (1) selection (maximum of four stars), (2) comparability (maximum of two stars), and (3) outcome (maximum of three stars). The selection domain includes “Representativeness of the exposed cohort”, “Selection of the non-exposed cohort”, “Ascertainment of exposure”, and “Demonstration that outcome of interest was not present at start of study” to evaluate the accuracy of the experimental group definition, the representativeness of the patient group, and the appropriateness of the control group. The comparability domain includes “Comparability of cohorts on the basis of the design or analysis controlled for confounders”. The outcome domain includes “Assessment of outcome”, “Was follow-up long enough for outcomes to occur”, and “Adequacy of follow-up of cohorts”, to evaluate the outcome evaluation method, evaluation timing, and accuracy. Then, the overall score was obtained by adding up the number of stars acquired across the three domains.

Two reviewers independently assessed the included six studies. Any unresolved disagreements between reviewers were resolved by a discussion with the third author.

### 2.6. Statistical Analysis

This meta-analysis investigated the outcomes of patients with hip fractures during the COVID-19 pandemic compared to before the pandemic. We calculated the pooled odds ratio (OR) with a 95% confidence interval (CI) using a random-effects model for mortality, presented as a forest plot. To minimize the influence of other variables as much as possible, the unadjusted OR value was used. When raw data were presented in the paper, the unadjusted odds ratio was calculated using the presented values. We estimated the inter-study inconsistency using the I^2^ test of the Higgins statistic to assess heterogeneity. Statistical heterogeneity was considered low if the I^2^ value was less than 25%, moderate if it was between 25 and 50%, high if it was between 50 and 75%, and very high if it was more than 75%. After obtaining the OR of outcome, the pooled effect size was estimated using the inverse variance weighted method.

We conducted planned subgroup analyses on extracted subgroup variables for the sample size, study facility, and study period. We performed a meta-analysis using statistical analysis software R (version 4.0.0, The R Foundation for Statistical Computing, Vienna, Austria) and packages “meta” (version 4.11-0) and “metaphor” (version 2.1-0). A *p*-value <0.05 was considered statistically significant. We assessed for publication bias using a funnel plot.

## 3. Results

In total, we identified 820 studies, with 611 studies remaining after we removed duplicates. We excluded 40 studies for irrelevance after assessing their titles and abstracts and retrieved the full texts of the 95 remaining relevant studies. We then excluded studies that had an irrelevant population (n = 39), irrelevant control (n = 3), irrelevant outcome (n = 11), or duplicated data (n = 2). Finally, we conducted a meta-analysis and systematic review, including 40 eligible studies with 17,753 patients with hip fractures, among which were 2280 patients with COVID-19 (12.8%) [11,20,21,22,23,24,25,26,27,28,29,30,31,32,33,34,35,36,37,38,39,40,41,42,43,44,45,46,47,48,49,50,51,52,53,54,55,56,57,58]. Except for Vialonga et al. (2020), all studies had a research period in 2020 [54]. Hall et al. (2022) reported the highest number of COVID-19 infections with 651 [36], followed by Rashid et al. (2022) with 517 infections [50]. Among the studies that reported the frequency of surgery, Barker et al. (2021) had the lowest rate of surgical treatment, with only 60% performed [21]. In all reported studies, the frequency of female patients was higher than that of male patients. Notably, in Jiménez-Telleria et al. (2020), the frequency of male patients was as low as 21% [37]. Table 1 shows the baseline characteristics of included studies, and Figure 1 presents the study selection flowchart.

### 3.1. Quality Assessment

We used the Newcastle–Ottawa Scale to evaluate article quality. All studies had four points in the selection domain and one point in outcome assessment, and 15 studies had an additional two points due to adjusting the confounding factors (Appendix A).

### 3.2. Overall and Regional Mortality of Patients with Hip Fractures during the COVID-19 Pandemic

The pooled 30-day mortality was 12.6% (95% CI 10.6–14.9%, I^2^ = 90%) during the COVID-19 pandemic for the 40 included studies involving 17,753 hip fractures (Figure 2). In the United Kingdom, the 30-day mortality of hip fractures during the pandemic was 14.3%; in Spain, it was 14.9%; in Italy, it was 9.6%; and in the United States (US), it was 6.2%.

### 3.3. Comparison of Pooled Mortality for Hip Fractures between Those with and without COVID-19 

We conducted a meta-analysis of patients with hip fractures between those with and without COVID-19 from the 40 included studies. Those with COVID-19 had significantly increased mortality compared to patients who did not have COVID-19 (OR, 7.10; 95% CI, 5.51–9.15; I^2^ = 57%; Figure 3)

### 3.4. Subgroup Analysis

We performed a subgroup analysis according to sample size (100 or more, or fewer than 100), study facility (single center or multi-center), and study period (the beginning of the pandemic or other periods). In all of the subgroup analyses, the 30-day mortality rates were significantly increased in those with hip fractures who had COVID-19 compared to those without COVID-19. Studies with fewer than 100 patients and in a single center were low-heterogeneity (12% and 19%) (Table 2).

### 3.5. Publication Bias

The funnel plot demonstrated symmetry, and we did not find publication bias in the included studies (Appendix A).

## 4. Discussion

As the COVID-19 pandemic unfolds, it is crucial to monitor the impact on healthcare systems and patient outcomes. The present study shows that the 30-day mortality rate for patients with hip fractures is significantly higher in those with COVID-19 infection. The 30-day mortality rate for patients with hip fractures during the COVID-19 pandemic was 12.6% in a meta-analysis, which is similar to previous studies (Clement et al., 2020; 13%) [15]. Factors such as hospital arrival delays, unequal allocation of medical resources, and limited surgical interventions to minimize the spread of infection must be accounted for to effectively manage patients with hip fractures during the pandemic [59,60]. Effective infection control measures, prioritizing high-risk patients, and proper communication and coordination between healthcare providers help ensure optimal outcomes for these patients during the pandemic [59,61,62].

The COVID-19 pandemic has had a profound impact on the mortality rates and healthcare systems of many countries. Previous meta-analyses mainly analyzed data from the UK, but this study aimed to evaluate mortalities by country [15,16]. The mortality rate in the UK was 14.3%, with Spain and Italy at 14.9% and 9.6%, respectively. In the US, four studies showed a mortality rate of 6.2%. The mortality rates of those infected with COVID-19 in the UK, Spain, Italy, and the US were 30.5%, 34.0%, 30.6%, and 20.8%, respectively, and the ORs were all significantly high (5.92, 5.18, 10.21, and 16.87, respectively). It is clear that COVID-19 infection increases the 30-day mortality of hip fractures.

In Europe, shortages of hospital beds, medical supplies, and staff early in the COVID-19 pandemic affected the mortality rate of patients with hip fractures. The 30-day mortality rates of patients with hip fractures were 1.4–10% in the 15 included studies of Giannoulis et al. (2016) [63], demonstrating that this 30-day mortality increased considerably during the pandemic. The UK’s initial strategy focused on herd immunity, but lockdowns were implemented as the situation worsened [64]. In Italy, strict lockdowns and other measures to halt the spread of the virus were necessary due to medical supply shortages and high numbers of patients [65,66]. In the US, the first COVID-19 wave was intense, leading to a widely criticized shortage of personal protective equipment and an overwhelmed healthcare system [67]. Conversely, South Korea, China, and Japan had lower mortality rates due to prompt pandemic management and a more robust healthcare system and infrastructure [68]. Pandemic responses and management varied widely across countries, depending on their healthcare system, economic stability, and government policies [64,67,68]. The most valuable lesson from Asia is the ability to prevent pandemics through improved hygiene and isolation of infectious individuals, as opposed to relying on severe economic shutdowns [69]. Several Asian countries have shown superlative results in suppressing the virus and keeping death rates per million incredibly low [69]. They learned from their experiences of the coronavirus that causes Severe Acute Respiratory Syndrome and favored rapid lockdowns or intensive mass testing and contact tracing without the need for a full-scale lockdown [70]. These different circumstances were reflected by the increase in the 30-day mortality rate of hip fractures.

However, the 30-day mortality of hip fractures in patients who did not have COVID-19 during the pandemic has not changed significantly. In this study, the mortality rate of patients without COVID-19 during the pandemic was similar to that of previous hip fracture deaths (Total: 6.13%, UK: 8.4%, Spain: 10.1%, Italy: 4.5%, and the US: 0.8%). After the first COVID-19 wave, the decrease in activity and the total number of patients indicated the possibility of increasing the emergency capacity for non-COVID-19 patients and providing efficient treatment. In addition, previous studies indicated that COVID-19 was a new mortality risk factor [27,35], and the high risk of COVID-19 infection during the pandemic increased the infection risk in patients with hip fractures. The direct effect of hip fracture on 30-day mortality is likely low.

This analysis revealed a prolonged research period, including the early stages of the pandemic, and included many patients. Several studies collected data from multiple institutions and countries through a registry, and Levitt et al. (2022) studied hip fracture mortality using nationwide data in the United States [41]. However, most of the included studies only encompassed data from the early stages of the pandemic. The pandemic is ongoing, so this analysis may not reflect the current medical system or account for the existing mortality rate of hip fractures due to COVID-19. Rashid et al. (2022) investigated the 30-day mortality of patients with hip fractures diagnosed with COVID-19 in the first and second waves and statistically confirmed a decrease in mortality in the second wave [50]. Although seven studies were conducted after the first wave, they included patients from the first wave. Future studies should examine the changes in mortality based on the pandemic period.

There are several limitations to this analysis. First, the study institutions’ medical resources were inadequately assessed. The medical resources available at each institution during the pandemic could not be determined, which could impact their response to patients with hip fractures. There was no information on whether patient accommodations were restricted or if examinations and treatments were limited. Second, it is difficult to generalize outcomes due to the limited study regions. This study is limited to Europe, with most data coming from the UK. Few studies were conducted in Asia, making it difficult to generalize these results. Third, most studies focused on the early pandemic stages, and only seven studies were conducted in the later pandemic stages. One case was classified by period, but the entire pandemic period should be analyzed at specific time points to account for the development of vaccines, therapeutic agents, and viral variants [63]. Finally, there were insufficient data on fracture types and comorbidities from the included studies. In some studies, the fracture type or underlying condition was reported, but lack of information prevented the collection of all relevant evidence. Future research requires a review based on a comprehensive investigation of this issue.

## 5. Conclusions

Patients with hip fractures who have COVID-19 have a significantly higher 30-day mortality rate than those without COVID-19. COVID-19 infection may have contributed to the increase in 30-day mortality for patients with hip fractures. The overall 30-day mortality rate for patients with hip fractures during the COVID-19 pandemic was 12.6%. Mortality rates varied by country, with Europe, including the UK and Spain, having the highest mortality rates.

Factors such as hospital arrival delays, unequal allocation of medical resources, and limited surgical interventions to minimize the spread of infection must be considered to effectively manage patients with hip fractures during the pandemic. Implementing effective infection control measures, prioritizing high-risk patients, and ensuring proper communication and coordination between healthcare providers can help achieve optimal outcomes for these patients. Future studies should systematically analyze these factors and track changes in mortality rates over time while investigating different pandemic periods.

## Figures and Tables

**Figure 1 jpm-13-00669-f001:**
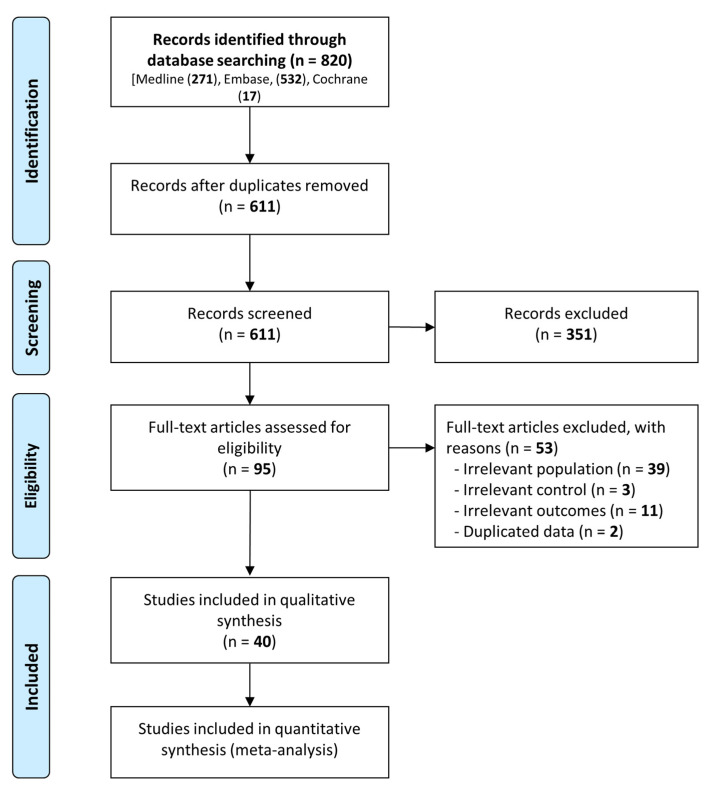
Flow diagram for the identification of relevant studies.

**Figure 2 jpm-13-00669-f002:**
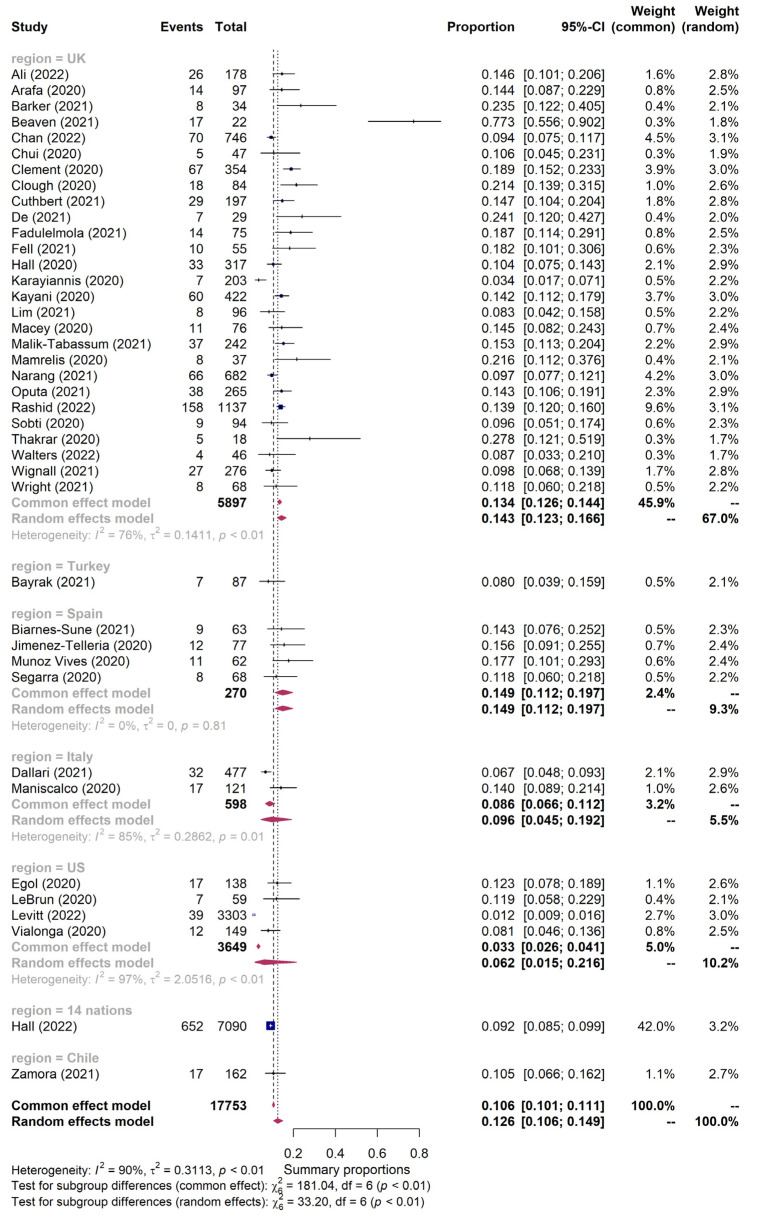
Forest plot for 30-day mortality of patients with hip fractures during the COVID-19 pandemic according to the study region [11,20,21,22,23,24,25,26,27,28,29,30,31,32,33,34,35,36,37,38,39,40,41,42,43,44,45,46,47,48,49,50,51,52,53,54,55,56,57,58].

**Figure 3 jpm-13-00669-f003:**
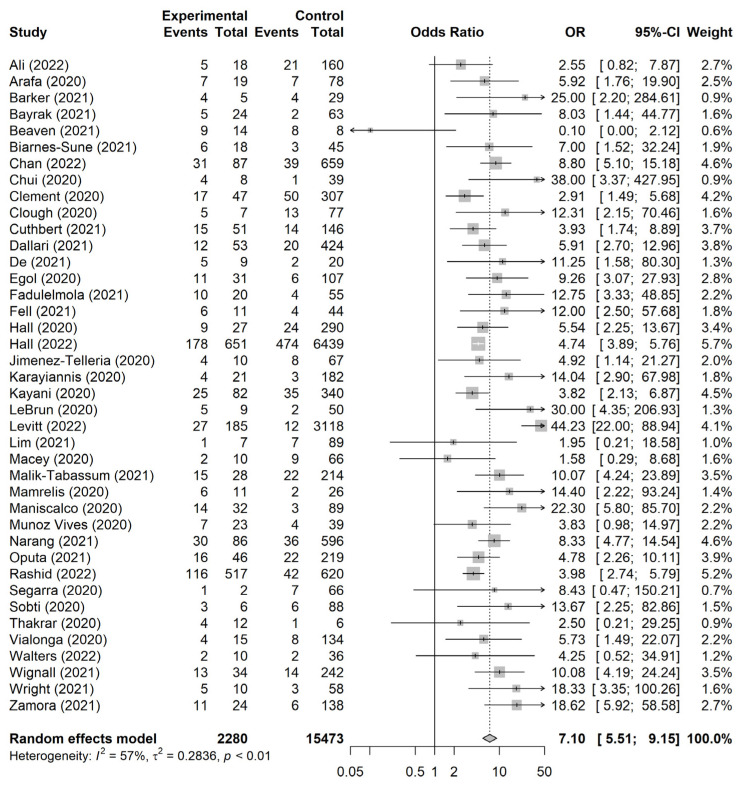
Forest plot for meta-analysis of patients with hip fractures with and without COVID-19 [11,20,21,22,23,24,25,26,27,28,29,30,31,32,33,34,35,36,37,38,39,40,41,42,43,44,45,46,47,48,49,50,51,52,53,54,55,56,57,58].

**Table 1 jpm-13-00669-t001:** Baseline characteristics of the included studies.

Study	Region	Period	Population		Age		Sex, Male,%		Rate of Operation		Length of Stay	
			COVID-19 +	COVID-19 −	+	-	+	-	+	-	+	-
Ali 2022 [20]	UK	March–June 2020	18	160	86	83	39	28	89	98	-	-
Arafa 2020 [11]	UK	1 March–31 May 2020	19	78	86 ± 8	83 ± 8	47	27	89	99	6 ± 1	5 ± 2
Barker 2021 [21]	UK	24 March–22 April 2020	5	61	-	-	-	-	60	100	-	-
Bayrak 2021 [22]	Turkey	April–November 2020	24	63	80 ± 15	79 ± 12	38	35	100	100	11 (9–16)	9 (7–11)
Beaven 2021 [23]	UK	28 March–25 May 2020	40	152	-	-	-	-	-	-	-	-
Biarnes-Sune 2021 [24]	Spain	11 March–24 April 2020	18	45	87 ± 7	85 ± 7	33	29	83	98	18± 9	11 ± 5
Chan 2022 [25]	UK	1 March–30 April 2020	87	659	86	83	38	28	-	-	-	-
Chui 2020 [26]	UK	31 March–29 April 2020	6	41	-	-	-	-	100	100	-	-
Clement 2020 [27]	UK	1 March–19 April 2020	68	1501	-	-	-	-	91	100	-	-
Clough 2020 [28]	UK	23 March–15 June 2020	7	77	85 ± 8	78 ± 11	57	36	100	100	17 ± 11	14 ± 6
Cuthbert 2021 [29]	UK	1 February–21 May 2020	51	146	79 ± 11	77 ± 13	51	37	98	99	23 (19–31)	9 (7–13)
Dallari 2021 [30]	Italy	8 March–4 May 2020	53	424	83 ± 1	81 ± 1	9	22	100	100	15 ± 2	11
De 2021 [31]	UK	1 March–15 May 2020	9	20	81	83	38	27	94	99	-	-
Egol 2020 [32]	US	1 February–15 April 2020	31	107	82 ± 10	83 ± 10	52	32	85	100	10 ± 5	5 ± 3
Fadulelmola 2021 [33]	UK	March–April 2020	20	55	83	84	35	27	95	96	-	-
Fell 2021 [34]	UK	23 March–12 May 2020	11	44	90 ± 8	86 ± 8	55	27	-	-	7 ± 7	5 ± 6
Hall 2020 [35]	UK	1 March–15 April 2020	27	290	84 ± 11	80 ± 11	52	32	93	96	-	-
Hall 2022 [36]	14 nations	1 March–31 May 2020	651	6439	84 ± 9	82 ± 11	37	29	60	62	17 ± 13	10 ± 8
Jiménez-Telleria 2020 [37]	Spain	9 March–15 April 2020	10	67	85 ± 7	85 ± 8	10	21	90	97	11 (7–11)	6 (5–8)
Karayiannis 2020 [38]	UK	18 March–27 April 2020	27	176	-	-	-	-	100	100	-	-
Kayani 2020 [39]	UK	1 February–20 April 2020	82	340	72 ± 10	73 ± 7	38	40	100	100	14 ± 5	7 ± 3
LeBrun 2020 [40]	US	20 March–25 April 2020	9	50	87 ± 8	85 ± 8	33	24	78	100	8 (4–13)	6 (3–10)
Levitt 2022 [41]	US	15 March–31 December 2020	185	3118	83 ± 8	82 ± 8	40	32	100	100	25 ± 6	25 ± 5
Lim 2021 [42]	UK	1 March–15 May 2020	7	89	88 ± 4	85 ± 9	14	28	100	95	30 ± 17	12 ± 7
Macey 2020 [43]	UK	20 March–25 April 2020	10	66	-	-	-	-	-	-	-	-
Malik-Tabassum 2021 [44]	UK	23 March–11 May 2020	28	214	87 ± 8	83 ± 8	32	30	96	99	16 ± 10	12 ± 8
Mamrelis 2020 [45]	UK	1 March–30 April 2020	11	26	84 ± 10	78 ± 10	38	30	73	89	-	-
Maniscalco 2020 [46]	Italy	22 February–18 April 2020	32	89	-	-	-	-	-	-	-	-
Munoz Vives2020 [47]	Spain	14 March–4 April 2020	23	39	-	-	-	-	-	-	-	-
Narang 2021 [48]	UK	1 March–30 April 2020	86	596	86	83	38	29	100	100	-	-
Oputa 2021 [49]	UK	5 March–5 April 2020	46	46	84 ± 7	82 ± 9	52	26	85	96	-	-
Rashid 2022 [50]	UK	23 March–31 December 2020	517	620	84	81 ± 10	35	31	99	97	24	13 ± 18
Segarra 2020 [51]	Spain	1 February–15 April 2020	2	66	88	82	50	32	100	944	7 ± 3	-
Sobti 2020 [52]	UK	1 March–31 May 2020	6	88	83	-	-	-	-	-	-	-
Thakrar 2020 [53]	UK	15 March–15 April 2020	12	6	-	-	-	-	-	-	-	-
Vialonga 2020 [54]	US	March 2020–March 2021	15	134	74 ± 21	78 ± 16	27	27	-	-	10.1 ± 6.2	7 ± 4
Walters 2022 [55]	UK	17 February–17 May 2020	10	36	-	-	-	-	-	-	-	-
Wignall 2021 [56]	UK	1 March–30 May 2020	34	242	85 ± 8	81 ± 12	41	37	-	-	18 ± 9	15 ± 11
Wright 2021 [57]	UK	11 March–30 April 2020	10	58	81 ± 11	-	-	-	90	98.3	17 ± 6	10 ± 9
Zamora 2021 [58]	Chile	15 March–30 August 2020	24	138	81 (75–88)	81 (77–89)	17	14	87	100	12.5 (7–23)	6.5 (4–11)

**Table 2 jpm-13-00669-t002:** Subgroup analysis for the 30-day mortality of hip fractures in patients with and without COVID-19.

Characteristics	The 30-Day Mortality			
	N	OR (95% CI)	*p*-Value for Heterogeneity	I^2^, %
All	40	7.10 (5.51–9.15)	<0.01	57
Population				
≥100	19	6.99 (5.03–9.71)	<0.01	73
<100	21	7.50 (5.09–11.03)	0.30	12
Facility				
Single center	18	6.16 (4.08–9.30)	0.22	19
Multi-center	22	7.54 (5.50–10.34)	<0.01	70
Period				
Beginning of the pandemic	33	6.34 (5.13–7.84)	0.04	32
Other periods	7	9.04 (3.96–20.68)	<0.01	86
Region				
United Kingdom	27	5.92 (4.63–7.58)	0.02	38
Spain	4	5.18 (2.32–11.56)	0.93	0
Italy	2	10.21 (2.84–36.73)	0.10	64
United States	4	16.87 (6.03–47.07)	0.02	70

## Data Availability

The datasets generated during the current study are available from the corresponding author on reasonable request.

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
