# Peer review of "Comparing the 30-Day Mortality for Hip Fractures in Patients with and without COVID-19: An Updated Meta-Analysis"

_jpm, 2023, doi:10.3390/jpm13040669_

Round 1
Reviewer 1 Report
The authors tried to describe a global overview of the impact of COVID-19 on the 30-day mortality of the hip fractures. However, there are some important issues that should be made clear.
1- Patients' characteristics of the included studies were not determined, e.g., type of fracture, comorbidities ...
2- Did the authors include only patients with operated hip fractures? What about those who were not fit for surgery or died before surgery???? This should be clear inn methods.
3- Although the author claimed that this is the first analysis to include countries other than UK, only 3 countries were included.
4- Authors mentioned that the quality of health systems in China, Korea ... is more efficient, but they did not mention how.
5- Authors always mentioned early COVID-19 period. Could you specify which wave or year????
Author Response
The authors tried to describe a global overview of the impact of COVID-19 on the 30-day mortality of the hip fractures. However, there are some important issues that should be made clear.
- Patients' characteristics of the included studies were not determined, e.g., type of fracture, comorbidities ...
Response:
Thank you for your insightful feedback. In order to conduct a meta-analysis, insufficient data on fracture type and individual comorbidities had to be collected from the included studies. Therefore, it was challenging to add to the baseline attribute. We will describe the contents in detail in the limitations section.
Revised:
Limitation in Discussion
~ Third, most studies focused on the early pandemic stages, and only seven studies were conducted in the later pandemic stages. One case was classified by period, but the entire pandemic period should be analyzed at specific time points to account for the development of vaccines, therapeutic agents, and viral variants [63]. Finally, there were insufficient data on fracture types and comorbidities from the included studies. In some studies, the fracture type or underlying condition was reported, but lack of information prevented the collection of all relevant evidence. Future research requires a review based on a comprehensive investigation of this issue.
- Did the authors include only patients with operated hip fractures? What about those who were not fit for surgery or died before surgery???? This should be clear inn methods.
Response:
Yes, all patients with hip fractures, regardless of whether or not they had surgery, were included in the inclusion criteria, hence Table 1 also includes the rate of patients who underwent surgery. Details will be provided in the Methods section. Thank you for your kind remarks.
Revised:
Methods
2.2. Eligibility criteria
~ The PICO questions were as follows: population (P) = all adult patients with hip fractures visiting the emergency room regardless conduction of operation; exposure (I) = COVID-19 infection; comparator (C) = non-infection; outcome (O) = 30-day mortality.
2.4. Study selection
~ We also excluded studies that 1) included patients aged < 18 years, 2) provided no comparisons or outcomes, and 3) were non-original articles. In addition, we included fracture patients with or without surgery. We subsequently reviewed the full text of potentially relevant articles that met the inclusion criteria.
- Although the author claimed that this is the first analysis to include countries other than UK, only 3 countries were included.
Response:
According to your comment, our wording is sufficiently misleading. The information about various countries was removed, and the introduction's objective was presented as having been updated since the previous meta-analysis.
Revised:
Introduction
We conducted an updated meta-analysis on the 30-day mortality of hip fractures for individuals with and without COVID-19 and also analyzed the 30-day mortality of hip fractures during the pandemic based on published casesfrom various countries.
- Authors mentioned that the quality of health systems in China, Korea ... is more efficient, but they did not mention how.
Response:
This is an essential point. We believe it should have been elaborated upon later in this sentence. Related information was added. Thanks.
Revised:
Third paragraph in discussion.
~ Pandemic responses and management varied widely across countries, depending on their healthcare system, economic stability, and government policies [64,67,68]. The most valuable lesson from Asia is the ability to prevent pandemics through improved hygiene and isolation of infectious individuals, as opposed to relying on severe economic shutdowns (69). Several Asians countries have shown superlative results in suppressing the virus and keeping death rates per million incredibly low (69). They learned from their experiences of the coronavirus that causes Severe Acute Respiratory Syndrome and favored rapid lockdowns or intensive mass testing and contact tracing without the need for a full-scale lockdown (70). These different circumstances were affected by the increase in the 30-day mortality rate of hip fractures.
- Authors always mentioned early COVID-19 period. Could you specify which wave or year????
Response:
In the first year (2020) following the declaration of the COVID-19 pandemic, there were about two peak periods in certain nations. In Rashid 2022, the population of the United Kingdom was examined by breaking the 2020 years into first and second waves (reference). Obviously, the peak period of infected patients following the announcement of a pandemic varies by country and location. However, given that vaccines were developed globally in the late 2020s and early 2021s, and afterwards as therapeutic agents, and since COVID-19 infections continue to occur intermittently, we believe it is reasonable to consider the first year (2020) to be an early period.
Reference: Rashid, F.; Hawkes, D.; Mahmood, A.; Harrison, W.J. Hip fracture mortality in patients co-infected with coronavirus disease 2019: a comparison of the first two waves of the United Kingdom pandemic during the pre-vaccine era. Int Orthop 2022, 46, 171-178, doi:10.1007/s00264-021-05269-x.
Reviewer 2 Report
I sincerely thank the editor for giving me an opportunity to review this article titled " Comparing the 30-day mortality for hip fractures in patients with and without COVID-19: An updated meta-analysis".
First of all I would like to congratulate authors for conducting this study and writing this meta-analysis. It highlights the 30-day mortality of hip fractures during the COVID-19 pandemic and assess mortality rates by country.
It is well written and the results are neatly prepared and presented. The common thread is clearly recognizable and the results show to attribute an even greater importance to the COVID-19 pandemic in the clinical routine.
Author Response
Thank you for your insightful remarks and for highlighting key points.
Round 2
Reviewer 1 Report
Authors adequately replied to all my comments. Many thanks.